# When News Topics Annoy—Exploring Issue Fatigue and Subsequent Information Avoidance and Extended Coping Strategies

Christina Schumann

Institute of Media and Communication Science, Department of Media Research and Political Communication, Faculty of Economic Sciences and Media, Technical University Ilmenau, 98684 Ilmenau, Germany; christina.schumann@tu-ilmenau.de

**Abstract:** This paper scrutinizes the phenomenon of issue fatigue and its consequences. Issue fatigue results from overexposure to a news topic that has been on the media's agenda for an extended period of time. Fatigued recipients become annoyed, and no longer wish to be exposed to the topic. Based on the findings of an explorative qualitative study, a quantitative online survey was conducted in Germany, Mexico, and Pakistan (N = 481). Using cluster analysis, we identified an emotional and a cognitive type of issue fatigue, and investigated how these types react. Both types of fatigued recipients avoided further news about the respective issue in traditional news media (= information avoidance). Differences were observed concerning the strategies to handle fatigue (= coping strategies): recipients of the emotional type posted about their fatigue in social media; recipients of the cognitive type turned to information in sources other than the mainstream news. Taking into account country-specific differences, we concluded that, generally, issue fatigue—via information avoidance—results in an uninformed citizenry. This can be a hurdle for the functioning of an established democracy or for the success of democratic transitions. Posting about issue fatigue, which was more frequent in Mexico and Pakistan, might 'infect' others, and intensify problems resulting from issue fatigue. Turning to alternative sources can be either beneficial or problematic for the development of a well-informed citizenry, depending on whether alternative sources provide reliable and truthful information.

**Keywords:** issue fatigue; information avoidance; alternative media; international comparison; Germany; Mexico; Pakistan

## 1. Introduction

Currently, many countries face problems that have been on the political agenda for a long time, such as the war against Ukraine, the coronavirus pandemic, migration flows, right-wing populism, and global warming. News outlets cover these contested issues intensively. Theories on the salience of news issues generally point to the positive effects of such intensive news coverage. Agenda-setting shows that recipients judge intensively covered news topics to be important (e.g., McCombs 2005). Consequently, recipients spend more time with, and attention to, such topics (Scheufele and Tewksbury 2007). They also engage more in related online searches, at least for some topics (Lee et al. 2016). Moreover, intensive coverage motivates recipients to process information more deeply (Ciuk and Yost 2016), to increase their knowledge about such topics (Matthes 2005), and even to engage in more political behavior, such as signing petitions or voting (Weaver 1991).

However, with the concept of issue fatigue, communication scholars have recently introduced a somewhat contrasting perspective: frequent and long-lasting news coverage on certain issues was found to provoke feelings of annoyance among recipients, paired with the thought that they no longer wished to hear or see any further related information (Gurr and Metag 2021; Kuhlmann et al. 2014; Metag and Arlt 2016; Schumann 2018). Examples of

issue fatigue topics include the coronavirus crisis (Wolling et al. 2021), the refugee crisis in Germany (Arlt et al. 2020) and 'Brexit' (Gurr and Metag 2021).

The potential outcomes of issue fatigue underline the relevance of the phenomenon: recipients experiencing issue fatigue tend to react—among other ways—by avoiding information. For example, they refuse further news exposure to the issue and/or related interpersonal discussions (Kuhlmann et al. 2014), even though the topic remains on both the political and media's agenda. Consequently, issue fatigue may provoke an inactive and uninformed citizenry. This is problematic, as a politically informed and participating citizenry is the foundation of a functioning democracy (Shehata 2016; de Vreese and Boomgaarden 2006). However, if broad segments of the population withdraw from current political issues, political decisions may be taken without the vigilance and legitimation of the citizens.

For this reason, we consider issue fatigue to be a relevant phenomenon that requires further scientific consideration. However, and as shown in the following paragraph, the concept remains vague, and empirical knowledge of the interplay between issue fatigue and subsequent information avoidance, or further reactions, is scarce. As such, the aim of this paper was twofold: firstly, we refined the actual concept of issue fatigue, and further explored its core elements in a qualitative approach. Here, qualitative data also indicated that we needed to extend our understanding of outcomes, as it was necessary to consider further reaction strategies in addition to 'mere' avoidance; secondly, we scrutinized under which conditions issue fatigue leads to various outcomes; for this, we used cluster and regression analysis based on a quantitative survey conducted in Germany, Mexico, and Pakistan.

## 2. Fatigue towards News Issues—Existing Perspectives

The idea that overexposure to a certain stimulus provokes negative consequences is a well-known concept in the field of advertising and campaign research. So-called 'wear-out effects' (e.g., Chen et al. 2016) show associations between stimulus repetition and decreased purchase intention, among other variables. Recently, for example, prolonged exposure to health campaigns was found to reduce or eliminate the amount of attention paid to advertising campaigns (Kim and So 2018; So et al. 2017).

Concerning news exposure, the idea of overexposure to an issue, and the potential negative consequences, is less established. Studies on issue attention cycles generally take into account the fact that public interest for issues wanes after a certain time span (Djerf-Pierre 2012; Downs 1972; Henry and Gordon 2001). The media is usually supposed to react to this decline in interest by shifting their focus to new or different issues (Downs 1972; see also research on 'media-hypes' by Vasterman 2005, 2018). However, these attempts do not consider the possibility that the media still pays a great deal of attention to certain issues when the public has already reached saturation point. Nevertheless, at least some studies suggest the existence of such a phenomenon. For example, (Capstick and Pidgeon 2014) found evidence for climate-change fatigue, which they defined as a feeling of boredom among recipients, and the perception that climate change is an outdated issue. (Beyer and Figenschou 2014) discovered human-interest fatigue among the Norwegian population, provoked by the unremitting news coverage about a young immigrant's deportation. Kinnick et al. (1996) investigated compassion fatigue, which is an emotional burnout or desensitization towards social problems, such as AIDS or homelessness, and showed that compassion fatigue was associated with avoidance of related news content. However, while the authors see avoidance as a predictor of burnout, we propose a reversed causality: in other words, the avoidance of related news content may be the outcome of compassion fatigue.

Taken together, the idea of overexposure, and its associated consequences, is recognized in different contexts, but the examples are still fragmented, and are not embedded in a coherent theoretical concept.

### 3. Issue Fatigue—A Definition

Health research describes fatigue as an experience that has cognitive and emotional dimensions (Ream and Richardson 1996; for an overview, see also Patel 2010). In the context of information consumption, emotional and/or cognitive responses due to repetition have been identified, e.g., from a prolonged exposure to health messages (Kim and So 2018; So et al. 2017), from repetition of persuasive statements in news articles (Koch 2017; Koch and Zerback 2013), or as a response to certain characteristics of political news, such as negativity (Schemer 2014). Similarly, psychology of human perception and information processing has shown that excessive repetition of stimuli—in our context, a news issue—provokes negative emotions (Bornstein 1989) and a decline in cognitive engagement (Berlyne 1970).

Taking the above into consideration, it is probably no coincidence that investigations into issue fatigue have shown that issue fatigue is comprised of negative emotional and cognitive responses toward a certain news issue (Kuhlmann et al. 2014; Metag and Arlt 2016). Based on these findings, Gurr and Metag (2021) defined issue fatigue as an "individual's negative state that emerges because of overexposure to an issue that news media cover extensively over a long period" (p. 1790). This definition shows that issue fatigue is temporal, and emerges over time.

Additionally, Gurr and Metag (2021) underlined that overexposure causes negative and emotional responses to the news issue. Schumann (2018) found that the following cognition–emotion combination formed the necessary condition for issue fatigue: the overexposure recipient no longer wanted to hear or see anything about the topic anymore (= cognition), and felt annoyed by the topic (= emotion). The cognition of no longer wanting to hear or see anything about a topic anymore is likely a manifestation of a decline in the issue-specific cognitive engagement (Matthes 2013) of a recipient. Over time, such declines result when a recipient perceives information as repetitive rather than newsworthy (Silvia 2006), as assumed for information on a frequently covered issue. Annoyance emerges in situations where situational factors hinder a person's goal (Roseman 1984). For example, in the context of media consumption, it was found that repetition of display advertisements causes annoyance to Internet users, as it interferes with the goal of information navigation (Todri et al. 2020). Interference with recipients' information goals is probably a key determinant of understanding why annoyance forms an inherent part of issue fatigue. In Kuhlmann et al.'s (2014) pilot study, recipients experiencing issue fatigue reported a mismatch between their informational needs and the informational value (Atkin 1973) of the news about the fatigue-inducing issue. This occurred either because excessive repetition of similar information about the issue made the recipients feel that the news only repeated what they already knew; or because overly detailed or overly complex coverage exceeded their capacity to process the information (information overload, Jackson and Farzaneh 2012).

The thought of no longer wanting to hear or see anything about a topic anymore, and annoyance, are the primary cognition and emotion, respectively, of issue fatigue. Without this specific combination, issue fatigue does not occur.

Primary emotion and cognition can (but do not have to) be paired with further emotions and/or cognitions (Gurr and Metag 2021). These are known as secondary emotions and cognitions and, together with the primary emotion and cognition, likely form different types of issue fatigue, and vary among different topics (Schumann 2018). For example, boredom could accompany issue fatigue toward coronavirus, due to stagnancy in the success in fighting the pandemic. In contrast, issue fatigue toward the war in Ukraine could coincide with sadness about the ongoing humanitarian crisis.

Despite these attempts at defining issue fatigue, existing perspectives still reveal shortcomings: firstly, while theoretically plausible, the assumption of primary and secondary emotion–cognition combinations has not yet been empirically tested; secondly, in regard to cognition, it is widely unknown as to which discrete dimensions come into play, as a systemat-

ical exploration is still lacking; lastly, it remains unclear which types of issue fatigue—namely, combinations of emotions and cognitions—exist, or whether they exist at all.

Moreover, to our knowledge, few studies have focused on the outcomes of issue fatigue or news saturation (Kinnick et al. 1996; Kuhlmann et al. 2014). These studies point to reactions that we identify as information avoidance. 'Information avoidance' is defined as deliberately shunning or delaying the acquisition of available information (Kim et al. 2020); often, it is used interchangeably with the term 'news avoidance' (Skovsgaard and Andersen 2020). However, this perspective overlooks the fact that citizens do not only inform themselves via the news, but that they also get information from interpersonal discussions. Thus, in our study, we differentiated between two levels of avoidance behavior: (a) further issue-related news; and (b) political talks with peers, family, and acquaintances. However, none of the existing studies have systematically explored the outcomes of issue fatigue. As such, further reaction types, other than information avoidance, could have been overlooked.

Finally, it remains unclear how different types of issue fatigue are associated with what kinds of reactions.

## 4. Research Questions and the Need for a De-Westernization of Perspectives

With regard to the aforementioned shortcomings, we ask:

- RQ1: What kinds of reactions do recipients develop when experiencing issue fatigue?
- RQ2: Which secondary emotions and cognitions of issue fatigue can be identified?
- RQ3.1: What types of issue fatigue exist?
- RQ3.2: Do different types of issue fatigue provoke different reactions?

The majority of research published in communications is limited to the industrialized Western world (Demeter 2019). Consequently, many world regions are simply not represented in the current discourse. This underrepresentation of research, from countries other than the global north, is one major point of criticism of the #CommunicationSoWhite movement (Ng et al. 2020). In line with this, scholars have demanded a so-called de-Westernization of communication research (Emmer and Kunst 2018), in order to understand the social constructs of non-Western countries (Chakravartty et al. 2018). This is particularly important as "theoretical models developed in the global north do not necessarily apply to other contexts. [...]. We know even less about whether, say, agenda setting or spiral of silence theories (or any of the others) would be the same all over the world if anyone ever tried to find it out" (Claussen 2020, p. 4, 7).[1]

The lack of global south representation is particularly problematic for internationally relevant research objects, such as issue fatigue (Schumann 2018). With our study, we aimed to contribute to this debate, and to overcome the myopic focus on results stemming only from the western world. We answered our research questions based on a study conducted in three countries: Germany, Mexico, and Pakistan. As we will show in the discussion part, reactions to issue fatigue may have different meanings in countries differing in democratization level or press freedom. Moreover, associations between issue fatigue and information avoidance or further reactions may not only depend on issue fatigue, but on cultural practices of media use (Hallin and Mancini 2004, 2012). Controlling for country-specific effects allows for the scrutinization of the theoretical stability of issue fatigue. As such, the three-country approach helps to develop the first step towards a generalizable theory of issue fatigue that is universally applicable despite external variables, such as cultural influences.

## 5. Qualitative Exploration of Reactions to Issue Fatigue as Well as Secondary Cognitions and Emotions

In our study, we combined a qualitative pre-study with a quantitative main study (online survey) in a mixed-methods design. We conducted our quantitative main study in three countries: Germany, Mexico, and Pakistan. To prepare the quantitative study, we realized an international qualitative pre-study. Both studies, qualitative interviews and quantitative

survey, were conducted together, with international graduate students participating in a research seminar on issue fatigue at a German university. The choice of countries we focused on in our study did not follow strict theoretical purposes. However, given the nationalities of the students, we were able to address the requirement of "looking inward" (Claussen 2020) with regard to these countries, which is crucial in operationalizing, conducting, interpreting, and understanding international research (Pathak-Shelat et al. 2015).

*5.1. Qualitative Pre-Study: Method*

We used a qualitative approach to explore reactions to (RQ1), as well as secondary emotions and cognitions relating to issue fatigue (RQ2) in an international context. For secondary emotions, we built on previous findings from a German master thesis (Kibke 2015).

The main purpose of the sampling technique was to set up an internationally diverse sample, with a focus on countries outwith the industrialized western world. In doing so, we wanted to explore and better understand the international meaning of issue fatigue, and to have a foundation for the international quantitative main study.

To this end, 15 semi-standardized written interviews in English, with participants from 13 countries, were conducted in November 2016. The participants were international students enrolled in an English-language Master's program at a German university. All of them had just recently (a maximum of 1 month ago) arrived in Germany. The countries of origin (in alphabetical order) were: Brazil, China, Colombia, Mexico, Morocco, Pakistan (two interviews), Peru, Portugal, Romania, South Korea, Turkey (two interviews), Vietnam, and Zimbabwe. The selection of these students, and their countries of origin, was determined by the opportunities we had in the research seminar to interview people from various world regions. Fortunately, we were able to realize a high share of interviewees from countries that are often underrepresented in international scientific discourse.

In the interviews, the participants were first asked if they remembered a recent news topic that they did not want to see, or hear about, any longer, and which made them feel annoyed; if they did, then they were asked to reflect on the emotions and cognitions provoked in them by topic confrontation in the news, and how they reacted to the topic ("When confronted with such a topic, what do you feel, think or do/ did you feel, think or do?").

The results, consisting of 46 pages of interview data, were analyzed by a qualitative content analysis, using MAXQDA 2018, following a semi-standardized approach. For this, we applied structuring qualitative content analysis (Mayring 2010) and defined the pre-set categories (= emotions, cognitions, and reactions) as maincodes in MAXQDA. The interview data were structured by similarities in respective sub-categories (= sub-codes in MAXQDA): for example, for the maincode 'cognition', the data revealed three sub-categories—'resignation', 'suspicion', and 'sarcasm' (see Table 1).

**Table 1.** Results from the qualitative exploration: reactions to issue fatigue, and secondary emotions and cognitions.

| Reactions to Issue Fatigue | | Refinement of Issue Fatigue | |
|---|---|---|---|
| **Information Avoidance** | **Extended Coping Strategies** | **Secondary Emotions** | **Secondary Cognitions** |
| • *traditional media:* active avoidance of issue-related news <br> • *social media:* unfollow sources <br> • *interpersonal discussions:* active avoidance of issue-related discussions | • *traditional media:* turn away from traditional media, and search for alternative perspectives <br> • *social media:* post issue fatigue <br> • *interpersonal discussions:* express issue fatigue | *High-arousal negative emotions:* <br> • frustration <br> • anger <br> • being upset <br> • disappointment <br> *Low-arousal negative emotions:* <br> • sadness <br> • boredom | • resignation <br> • suspicion <br> • sarcasm |

In the following, we first turn to the results on reactions triggered by issue fatigue, before presenting the results about secondary emotions and cognitions.

*5.2. Results on Reactions to Issue Fatigue: Information Avoidance and Extended Coping*

The data suggested two broader reaction types (see Table 1): 'information avoidance' and 'extended coping strategies', with the latter not having been found in previous work.

5.2.1. Information Avoidance

In line with previous findings, recipients of issue fatigue actively and consciously avoided further news about the issue. For traditional media, participants reported changing the TV channel or radio station, closing a news webpage, or ceasing to read an article in a newspaper. For example, the interviewee from Mexico said: "I stopped watching the news and when the radio mentioned it, I just turned it off or changed to another station." In social media, interviewees indicated that they unfollowed sources spreading information about the topic, such as in this example from the Romanian interview: "I have even unsubscribed from the Facebook pages which were still providing this topic." Moreover, participants reported ignoring related interpersonal discussions.

5.2.2. Extended Coping Strategies

When reacting with extended coping, recipients actively dealt with their issue fatigue. For traditional media, participants reported not only cutting themselves off from the news stream presented in traditional news (= information avoidance), but at the same time, searching actively for information about the topic in alternative, non-journalistic sources, such as YouTube channels or online blogs: "I look for more independent and skeptic thoughts on the topic" (Turkish interviewee). For social media and interpersonal discussions, participants reported using them as an outlet to express their fatigue. The interviewee from China explained: "Interpersonal communication is a way to vent one's annoyance and disagreement. [ . . . ] I like to express my annoyance online."

*5.3. Results on Secondary Emotions and Cognitions*

5.3.1. Secondary Emotions

Just as in the master thesis (Kibke 2015), we found evidence for an increase in both high- and low-arousal negative emotions, as classified in the circumplex model of affect (Russell 1980), such as anger or frustration (high arousal) or boredom and sadness (low arousal). Giving an example for high-arousal emotions, a Pakistani said: "I grew anger towards this topic". For low-arousal emotions, the Brazilian interviewee, for example, said: "But after some weeks seeing the topic in every news channel, the topic started to bother and created a sensation of boredom."

5.3.2. Secondary Cognitions

Participants also reported cognitively resigning, due to issue overexposure, and becoming indifferent toward it, as stated here: "At the end, I felt resigned" (Colombian interviewee). Additionally, interviewees mentioned thoughts of suspicion about the intensive news coverage of one specific issue, and wondering whether its intent was to distract from other issues, as shown in this example: "I wondered myself who is behind all this and for what purpose?" (Moroccan interview). Lastly, participants reacted with sarcastic and ironic consideration of the fatigue-inducing issue, e.g., by making jokes about the issue ("Now people started joking" [ . . . ], Pakistani interview).

5.3.3. Theoretical Considerations of the Findings on Secondary Emotions and Cognitions

From a theoretical standpoint, certain combinations of primary emotion and cognition with secondary emotions and cognitions may trigger more intense reactions than others. For example, boredom is seen as a strong driver for people to avoid further contact with a stimulus (e.g., Atkin 1973, 1985; Hill and Perkins 1985; Mikulas and Vodanovich 1993). As

such, issue fatigue combined with boredom could trigger information avoidance. The same could hold true for negative high-arousal emotions. In this context, avoiding information is a strategy for escaping negative experiences (Ytre-Arne and Moe 2021) and maintaining individual wellbeing (Boukes and Vliegenthart 2017), which is also underlined in the information utility approach (Atkin 1985; Brashers 2001).

By contrast, a sarcastic consideration still points to a certain level of cognitive engagement with the issue. In the context of political satire, sarcasm or irony are defined as forms of criticism or assault (e.g., Jones 2005). Moreover, sarcasm is usually perceived as entertaining (Simpson 2003), which could make issue fatigue (more) bearable and 'help' people keep up with an issue, instead of completely avoiding it.

However, for the other identified emotions and cognitions, it was difficult to assume theoretically how they might relate to information avoidance or extended coping. Moreover, issue fatigue could be paired with more than just one single secondary emotion or cognition. As such, we did not formulate discrete hypotheses.

## 6. Quantitative Exploration of Different Types of Issue Fatigue and Their Associations with Information Avoidance and Extended Coping

In order to answer RQ3, we conducted an online survey in Germany, Mexico, and Pakistan, in May and June 2017, with citizens residing in those countries. As stated above, the project was realized in a research seminar with international graduate students, in a public German university. The students were also in charge of the data collection. The choice of countries for our study did not follow theoretical purposes, but we used the insider-knowledge of the students about their home countries, and their close connections to the people, for data collection, in order to be "looking inward" (Claussen 2020), and to realize an international sample (Pathak-Shelat et al. 2015).

Participants in all countries were recruited via personal contacts, university mailing lists, and Facebook groups. All participants were required to indicate their level of issue fatigue for two current news topics, before being asked about further emotional and cognitive considerations, as well as information avoidance and extended coping. To this end, we first identified potentially fatigue-inducing topics in all countries.

### 6.1. Identification of Issue Fatigue Topics

Generally, an 'issue' comprises events that are perceived to belong together (Kepplinger 2001). In that sense, 'issues' are "cognitive sense making tools serving as overarching categories or bins [ . . . ] into which news stories and the events they cover can be sorted" (Geiß 2018, p. 85). Recipients assign a label, such as 'coronavirus', to these overarching categories, under which they classify perceived sub-issues (Eichhorn 1996; Geiß 2018). For issue identification, we conducted a pre-study (online survey) in each country, that aimed to identify such labels. Firstly, as an open-ended question, we asked participants to identify their potential fatigue issues (if they had any), and also to describe them. The survey was distributed in April 2017, via Facebook and personal contacts of the students. In total, N = 227 participants were acquired (Germany: n = 39; Mexico: n = 133; and Pakistan: n = 55). Of these, 82% (n = 186) mentioned at least one fatigue issue (Germany: 77%, n = 30; Mexico: 81%, n = 108; Pakistan: 87%, n = 48). For issue identification, we first coded the aforementioned issues by commonalities (e.g., "attacks in Syria" and "war in Syria" were coded as issue 'Syria'). Afterwards, we chose two topics based on the following criteria: firstly, that the probability that the topic would remain on the media's agenda for the duration of the main survey should be high; secondly, that topics should be identical for all countries, as this allowed controlling for country-to-country influences in the regression models; thirdly, that the topics should have been mentioned by several participants; fourthly, that topics should not be too abstract—for example, mentions such as "politics in general" probably did not refer to issue fatigue, but rather to a more general political malaise (for an overview, see Maurer 2003; Wolling 1999); fifthly, topics should be politically, economically, and/or socially relevant; as such, we excluded mentions of non-political or 'tabloid' issues, such

as scandals, crime, etc., which were relatively numerous (N = 60 for the whole sample). Applying these criteria, we first chose the topic 'Trump', with n = 52 mentions in total (= 28%; Germany: n = 15; Mexico: n = 35; Pakistan: n = 2). As Donald Trump is not an issue, but rather a person, the open-ended questions about the topic description showed what people had in mind when they mentioned Trump as a topic: his (perceived) aggressive leadership style, his (perceived) ignorance of other countries, and the considerable media attention that is paid to any of his actions. 'War in Syria' was chosen as the second topic, with n = 5 mentions in total (Germany: n = 1; Mexico: n = 3; Pakistan: n = 1).

### 6.2. Item Development and Measures

The operationalization of issue fatigue, as well as information avoidance, was based on the measures from Kuhlmann et al. (2014). Secondary emotions were taken from the circumplex model of affect (Russell 1980). For secondary cognitions, as well as extended coping, we developed new items, which closely resembled the expressions used by the participants of the qualitative study.

The first draft of the questionnaire (English language) was tested in a quantitative pre-test with English-speaking graduate students at a public German university (N = 74). Scales comprised of more than three items (= emotions and secondary cognitions) were further analyzed, using exploratory factor analyses to examine the stability of scale dimensions. After finalizing the English version, native speaking students translated the survey from English into German and Spanish.

### 6.2.1. Issue Fatigue and Secondary Emotions and Cognitions

We measured topic fatigue by combining primary emotion and cognition into the following questions: "Currently: When you think about your own thoughts and feelings when being confronted with the two aforementioned topics in the media—do you have the feeling that one or both of these topics annoy you somehow? Does one or both of these topics make you think: I do not want to hear nor see anything about this anymore!?" Afterwards, participants indicated their level of fatigue on a 5-point Likert scale from "not at all annoyed" to "very annoyed" for both topics. The mean level of issue fatigue for 'Trump' was $M = 3.1$ ($SD = 1.3$), and for 'Syria' $M = 1.8$ ($SD = 1.3$).

For secondary emotions and cognitions, we used 5-point Likert scales ranging from "not at all" to "very strong". We used one item each for the emotions, and two items each for the cognitions (see Table 2).

**Table 2.** Indices and single items for information avoidance, extended coping, and secondary emotions and cognitions.

| **Information Avoidance** |
|---|
| (scale from '(1) = never' to '(5) = very often') |
| *Active avoidance of news in traditional media* (index of four items, $M = 2.3$, $SD = 1.0$, $n = 314$ [1]) |
| • Television: switch to another station ($M = 2.5$, $SD = 1.1$, $n = 65$) |
| • Print news: stop reading an article ($M = 2.4$, $SD = 1.0$, $n = 120$) |
| • Online news: leave the page ($M = 2.2$, $SD = 1.0$, $n = 155$) |
| • Radio: switch to another station (M = 2.1, SD = 1.2, n = 30) |
| *Active avoidance in social media* |
| • Unfollow accounts that share related content ($M = 1.9$, $SD = 1.2$, $n = 226$) |
| *Active avoidance of interpersonal discussions* |
| • Ignore the discussion ($M = 2.7$, $SD = 1.3$, $n = 463$) |

[1] The total n is different to the sum of the n of the single items; as participants filled in the survey for information avoidance for the two media types (max.) they got information about the respective topic 'often' or 'very often'

**Table 2.** *Cont.*

| **Extended Coping** |
| --- |

(scale from '(1) = never' to '(5) = very often')
*Traditional media*

- Turn away from traditional news and search for alternative perspectives (*M* = 2.9, *SD* = 1.2, *n* = 464)

*Social media*

- Post a comment or tweet that showed your annoyance (*M* = 2.5, *SD* = 1.4, *n* = 264)

*Interpersonal discussions*

- Tell others that the topic is bothering you (*M* = 2.2, *SD* = 1.2, *n* = 461)

| **Secondary Emotions** |
| --- |

(scale from '(1) = not at all' to '(5) = very strong')
*Negative emotions of high arousal* (index of four items; *M* = 3.1, *SD* = 1.1, alpha =.82)

- Anger (*M* = 3.0, *SD* = 1.3, factor loading: 0.84)
- Upset (*M* = 3.2, *SD* = 1.4, factor loading: 0.87)
- Frustration (*M* = 3.2, *SD* = 1.4, factor loading: 0.87)
- Disappointment (*M* = 3.2, *SD* = 1.4, factor loading: 0.60)

*Sadness* (single item, *M* = 3.1, *SD* = 1.4, factor loading: 0.92)
*Boredom* (single item, *M* = 2.3, *SD* = 1.3, factor loading: 1.0)

Principal component analysis with varimax-rotation, 77 percent variance explained, KMO: 0.844
*Comment:* The circumplex model of affect classifies sadness as low-arousal emotion. Moreover, (reduced) sadness played a crucial role in existing studies on compassion fatigue. As such, we rejected the initial factor solution in which sadness loaded on the high-arousal emotion factor, and opted for a three-factor solution.

| **Secondary Cognitions** |
| --- |

(scale from '(1) = never' to '(5) = very often')
*Suspicion* (index of two items; *M* = 3.0; *SD* = 1.1, alpha = 0.67)

- think the topic is intentionally launched by certain interest groups (*M* = 3.3; *SD* = 1.2, factor loading: 0.88)
- think the topic is used to distract people from other important topics (*M* = 2.8; *SD* = 1.4, factor loading: 0.82)

*Sarcasm* (index of two items: *M* = 2.3, *SD* = 1.3, alpha = 0.88)

- make sarcastic comments about the topic (*M* = 2.3, *SD* = 1.4, factor loading: 0.93)
- make jokes about the topic (*M* = 2.2, *SD* = 1.3, factor loading: 0.93)

*Resignation* (single item)
think that arguing about the topic is pointless (*M* = 2.9; *SD* = 1.2. factor loading: 0.97)

Principal component analysis with varimax-rotation, 85 percent variance explained, KMO: 0.61

Note: We dropped the second item operationalized for resignation, as it showed double loadings on the other factors.

### 6.2.2. Information Avoidance and Extended Coping

We used 5-point Likert scales for measuring information avoidance and extended coping, ranging from '1 = never' to '5 = very often' (see Table 2).

### 6.2.3. Controls

We controlled for (dis)interest, as information avoidance may be highly dependent on the personal interest a person has in a topic (Silvia 2006). Moreover, as women seem to be more prone to developing issue fatigue (Kuhlmann et al. 2014), we controlled for gender. Finally, and as indicated above, we used 'country' as a control variable, as media usage or avoidance patterns may depend on cultural habits (Hallin and Mancini 2004, 2012).

### *6.3. Procedure*

#### 6.3.1. Structure of the Questionnaire

Participants indicated their level of issue fatigue for both topics. Afterwards, we assigned one issue randomly to each participant. For this specific topic, participants answered questions about secondary emotions and cognitions, as well as information avoidance and coping strategies. This random assignment was used to prevent participants

from having to answer the same questions several times and likely abort the survey on that account. Information avoidance, as well as coping strategies, were only asked about in regard to the two most-used media types, which we analyzed in a different question beforehand.

6.3.2. Sample

In total, N = 481 completed the survey with n = 240 answering "Trump" and n = 241 "war in Syria". With a mean age of 27 years (SD = 10.0, MIN = 17, MAX = 70), participants belonged mostly to the younger parts of society, with the youngest participant being 17 years old. Gender distribution was nearly equal, with n = 240 (50%) female and n = 229 (48%) male participants (n = 12, 2% indicated third gender or preferred not to answer). Formal education level was very high, with only 2% below a general matriculation standard (n = 15), while 30% (n = 144) had achieved a general matriculation standard or even a university degree (68%, n = 327). With regard to country of origin, Pakistanis were slightly underrepresented, as n = 127 participants (26%) came from Pakistan, while n = 177 (37%) came from both Germany and Mexico.

*6.4. Analytical Strategy*

6.4.1. Indices for Information Avoidance and Extended Coping

For further analysis, we prepared our data as follows: we built an index for information avoidance in traditional media. For information avoidance on social media and in interpersonal discussions, as well as extended coping, we used the single items operationalized in the survey (see Table 2).

6.4.2. Identification of Types of Issue Fatigue with Secondary Emotions and Cognitions

In order to identify people who had specific combinations of primary with secondary emotions and cognitions, we built indices based on factor analysis (see Table 2).

After the indices, we used hierarchical cluster analysis with the single-linkage method in order to exclude outliers, and subsequently Ward's method with the Elbow-criteria, for cluster identification. We conducted one cluster analysis per issue. The results revealed four clusters for each of the respective issues (see Table 3). To study the association between cluster affiliation and information avoidance or extended coping, we followed the procedure for using categorical variables in regression analysis (Kassambara 2018). We defined one cluster as the baseline category, and recoded the other cluster affiliations into categorical variables with two levels: 'belonging to cluster (1)' or 'not belonging to cluster (0)', and used them as predictors in the regression model. Consequently, the regression coefficients of the predictors had to be interpreted relative to the baseline category. For 'country' (the control variable), we used the same procedure.

**Table 3.** Types of issue fatigue: combinations of primary and secondary emotions and cognitions (cluster analysis).

| Clusters | Trump | | | | War in Syria | | | |
|---|---|---|---|---|---|---|---|---|
| | **The Apathetic (Baseline)** | **The Fatigued-Suspicious** | **The Fatigued-Suffering-From-All** | **The Negative-Emotional** | **The Apathetic (Baseline)** | **The Fatigued-Suspicious** | **The Compassionate** | **The Cognitive-Emotional** |
| Short description | Relatively low on all cognitions and emotions | The cognitive type of issue fatigue with relatively high suspicion and resignation | Issue fatigue paired with all other dimensions | No issue fatigue, but emotional involvement | Relatively low on all cognitions and emotions | The cognitive type of issue fatigue with relatively high suspicion and resignation | No issue fatigue, but compassion | No issue fatigue, but emotional and cognitive involvement |
| N (percent of sample) | 64 (29%) | 72 (32%) | 36 (16%) | 50 (22%) | 43 (22%) | 39 (20%) | 93 (47%) | 24 (12%) |
| Issue fatigue | 1.8 | 4.3 | 3.7 | 2.6 | 1.9 | 3.5 | 1.2 | 1.3 |
| High-arousal negative emotions | 2.4 | 2.5 | 4.1 | 3.5 | 2.4 | 3.4 | 3.9 | 3.6 |
| Sadness | 1.9 | 1.6 | 3.9 | 3.6 | 2.3 | 3.5 | 4.6 | 4.0 |
| Boredom | 1.5 | 2.9 | 3.2 | 3.5 | 1.6 | 2.4 | 1.4 | 3.5 |
| Suspicion | 3.0 | 3.7 | 3.4 | 3.3 | 2.9 | 3.1 | 2.4 | 3.2 |
| Sarcasm | 3.0 | 2.8 | 4.1 | 2.6 | 1.7 | 1.8 | 1.2 | 2.2 |
| Resignation | 2.8 | 3.5 | 3.7 | 2.6 | 2.1 | 3.5 | 2.3 | 3.0 |

## 7. Results

### 7.1. RQ3.1: Types of Issue Fatigue—Combinations of Primary and Secondary Emotions and Cognitions

Both cluster analyses (Table 3) revealed one group of participants who were predominantly indifferent to the respective topic, as they scored relatively low on all emotions and cognitions (other than a mild tendency toward sarcasm for Trump). We labeled these as the apathetic, and used them as a baseline cluster in the regression analysis.

The second cluster appeared for both issues as well. Here, the level of issue fatigue was highest, compared to all other clusters. Moreover, the participants in this cluster scored relatively highly on two of the secondary cognitions—namely, suspicion and resignation. As such, issue fatigue was paired with strong further cognitive considerations. As suspicion was more prevalent, we labeled this cluster 'the fatigued-suspicious'.

Cluster 3 and 4 varied between the two issues. For Trump, we had one additional cluster scoring highly on issue fatigue: cluster 3. Furthermore, the participants showed high levels of all other dimensions, pointing to a highly emotional and cognitive engagement with the issue. Thus, we named the respective participants 'the fatigued-suffering-from-all'. Lastly, with the negative-emotional, we found one cluster with individuals responding with negative emotions exclusively, without 'suffering' from issue fatigue or showing deeper cognitive consideration.

For the war in Syria, clusters 3 and 4 did not show evidence of issue fatigue. Instead, cluster 3 was comprised of individuals with negative, high-arousal emotions and sadness but no boredom, pointing to a high level of empathy. Referring to compassion fatigue, as introduced in the theory section (Kinnick et al. 1996), we named this cluster 'the compassionate'. Interestingly, cluster 4 showed relatively high scores on all secondary emotions and cognitions, but not on issue fatigue. We labeled this cluster 'the cognitive-emotional'.

### 7.2. RQ3.2: Types of Issue Fatigue and Their Relation to Information Avoidance and Extended Coping

In the following, we first present the associations between types of issue fatigue and information avoidance (Table 4) and afterwards turn to the results for extended coping (Table 5). In each case, we first discuss the effects of cluster affiliation before turning to the control variables. After each table, we briefly summarize the main findings.

**Table 4.** Associations between issue fatigue clusters and information avoidance—multiple linear regression analysis with categorical variables and baseline category.

| | Information Avoidance | | | | | |
|---|---|---|---|---|---|---|
| | **Traditional Media** | | **Social Media (Un-Follow Sources)** | | **Interpersonal Discussions** | |
| | **Trump** | **Syria** | **Trump** | **Syria** | **Trump** | **Syria** |
| *Trump Cluster (baseline cluster: the apathetic)* | | | | | | |
| The fatigued-suspicious | 0.21 *[1] | | 0.05 | | 0.22 ** | |
| The fatigued-suffering-from all | 0.22 ** | | 0.35 *** | | 0.14 * | |
| The negative-emotional | 0.14 ' | | 0.09 | | 0.20 ** | |
| *Syria Cluster (baseline cluster: the apathetic)* | | | | | | |
| The fatigued-suspicious | | 0.27 ** | | −0.00 | | 0.10 |
| The compassionate | | −0.10 | | 0.36 * | | −0.09 |
| The cognitive-emotional | | 0.13 | | 0.08 | | 0.07 |
| *Controls (baseline for country: Germany)* | | | | | | |
| Disinterest | 0.21 ** | 0.28 ** | 0.02 | −0.15 | 0.25 *** | 0.15 * |
| Gender (m) | −0.07 | −0.11 | −0.06 | 0.10 | −0.05 | 0.04 |
| Mexico | 0.13 ' | −0.04 | 0.05 | 0.42 * | 0.07 | 0.02 |
| Pakistan | 0.29 *** | −0.04 | 0.16' | 0.48 * | 0.12 | −0.10 |
| $R^2$ | 20% | 22% | 12% | 23% | 14% | 7% |
| $R^2$ (adj.) | 17% | 17% | 9% | 13% | 11% | 4% |
| F | 6.31 | 4.35 | 1.01 | 2.29 | 4.90 | 2.11 |
| df | 7 | 7 | 7 | 7 | 7 | 7 |
| N | 184 | 116 | 154 | 63 | 223 | 219 |

[1] Reading example: In comparison to the apathetic, the 'fatigued-suspicious' more often avoid news about Trump in traditional media. ' $p < 0.10$. * $p < 0.05$. ** $p < 0.01$. *** $p < 0.001$.

**Table 5.** The associations between issue fatigue clusters and extended coping—multiple linear regression analysis with categorical variables and baseline category.

| | Extended Coping Strategies | | | | | |
| --- | --- | --- | --- | --- | --- | --- |
| | Turn Away from Traditional Media and Search for Alternative Perspectives | | Post Issue Fatigue in Social Media | | Express Issue Fatigue in Interpersonal Discussions | |
| | Trump | Syria | Trump | Syria | Trump | Syria |
| *Trump Cluster (baseline cluster: the apathetic)* | | | | | | |
| The fatigued-suspicious | 0.13 ′ | | 0.00 | | 0.13 ′ | |
| The fatigued-suffering-from all | 0.01 | | 0.25 ** | | 0.25 ** | |
| The negative-emotional | 0.05 | | 0.08 | | 0.02 | |
| *Syria Cluster (baseline cluster: the apathetic)* | | | | | | |
| The fatigued-suspicious | | 0.23 ** | | −0.11 | | 0.09 |
| The compassionate | | 0.02 | | 0.26 * | | −0.11 |
| The cognitive-emotional | | 0.11 | | −0.13 | | −0.01 |
| *Controls (baseline for country: Germany)* | | | | | | |
| Disinterest | −0.16 * | −0.20 ** | −0.13 | −0.06 | −0.20 ** | 0.02 |
| Gender (m) | 0.19 ** | 0.08 | −0.05 | 0.09 | −0.01 | 0.07 |
| Mexico | 0.09 | 0.17 * | 0.29 ** | −0.01 | 0.20 ** | 0.24 ** |
| Pakistan | −0.13 | −0.18 * | 0.38 * | −0.19 | 0.09 | 0.23 ** |
| $R^2$ | 20% | 22% | 20% | 18% | 12% | 10% |
| $R^2$ (adj.) | 17% | 17% | 15% | 11% | 9% | 6% |
| F | 3.38 | 7.28 | 5.01 | 2.69 | 4.01 | 3.2 |
| df | 7 | 7 | 7 | 7 | 7 | 7 |
| N | 220 | 221 | 159 | 94 | 222 | 219 |

Reading example: In comparison to the apathetic, the 'fatigued-suspicious' more often avoid news about Trump in traditional media. ′ $p < 0.10$. * $p < 0.05$. ** $p < 0.01$.

### 7.2.1. Types of Issue Fatigue and Information Avoidance

Table 4 shows the associations between cluster affiliation and information avoidance in traditional media, social media, and interpersonal discussions.

Belonging to the fatigued-suspicious showed positive associations with information avoidance in traditional media (both topics) and interpersonal discussions (Trump only).

Similarly, the fatigued-suffering-from-all (Trump) avoided news in the traditional media and in interpersonal discussions. Additionally, they set sources in social media to 'unfollow'.

Clusters that did not comprise issue fatigue showed only two effects: the negative-emotional tended to avoid discussions about Trump, and the compassionate unfollowed sources that shared content about Syria.

For controls, we found disinterest to be a predictor of information avoidance in traditional media and interpersonal discussions. Concerning country, Pakistani and Mexicans more often avoided news about Trump in traditional media, and also more often unfollowed sources posting information about Syria than Germans. Effect sizes here were comparably high.

As an *interim conclusion*, we summarize: both types of issue fatigue led to subsequent information avoidance in traditional news and interpersonal discussions. Additionally, the fatigued-suffering-from-all also reacted in social media by deleting topic-sharing sources from their news stream. These effects were present even though we controlled for disinterest and country influences, indicating a very stable phenomenon.

Furthermore, regressions showed more and stronger coefficients between information avoidance and clusters containing issue fatigue than between information avoidance and clusters not containing issue fatigue.

Considering nationality, both Pakistani and Mexicans showed tendencies to turn away from traditional media and 'adjust' their social media news amalgam more often than Germans.

### 7.2.2. Types of Issue Fatigue and Extended Coping

Table 5 shows the associations between types of issue fatigue and extended coping.

The fatigue-suspicious tended to turn away from the discourse in traditional media, and to search for alternative perspectives on the topic outside of the mainstream news

(both topics). For Trump, there was an additional slight tendency to use interpersonal discussions to express their fatigue.

By contrast, the fatigued-suffering-from-all did not search for alternatives, but did clearly express their fatigue regarding Trump, in both social media and interpersonal discussions.

For the non-fatigue clusters, only one effect was found for the compassionate who, surprisingly, indicated that they posted about their fatigue regarding Syria on social media.

Concerning controls, disinterest 'prevented' recipients from actively searching for alternative information about Trump and Syria and expressing their fatigue in interpersonal discussions. Moreover, we found a gender effect, as men more often searched for alternative perspectives on Trump than women. Regarding country, Mexicans searched more often for an alternative perspective on Syria than Germans, while Pakistanis did so less often. Concerning posting and expressing fatigue, Mexicans and Pakistanis showed similar reactions, as they both conveyed their fatigue about Trump on social media, and expressed fatigue in interpersonal discussions (Syria: both; Trump: Mexicans only) more often than Germans.

We summarize the extended coping strategies: expressing fatigue—either online or offline—was a strategy shown by the fatigued-suffering-from-all, while the fatigued-suspicious searched for alternative perspectives on the issue, that were not part of the mainstream coverage. Additionally, expressing fatigue also seemed to be more a Mexican and Pakistani reaction than a German one.

## 8. Discussion

The results of this study broaden existing perspectives on issue fatigue: firstly, for extended coping, our paper sheds light on outcomes of issue fatigue not detected in previous research; secondly, with the fatigued-suspicious and the fatigued-suffering-from-all, we identified two different types of issue fatigue, by applying the theoretical assumption of secondary emotions and cognitions. The first of these appeared for both topics, and had a clear cognitive component, as primary emotion and cognition go hand in hand with thoughts of suspicion and resignation. The second appeared solely for Trump, and showed high cognitive and emotional involvement.

The results from regression analysis revealed manifold associations of both types of issue fatigue with information avoidance and extended coping. The effects were present even though we controlled for disinterest and country influences, indicating that issue fatigue is a stable and robust phenomenon.

Likewise, the comparison of the regressions between clusters containing issue fatigue and clusters not containing issue fatigue showed that the fatigue clusters had more and stronger effects on information avoidance and extended coping. In other words, information avoidance or extended coping were not triggered by negative emotions or cognitions per se, but only when issue fatigue was present. That is, participants did not avoid information when they were 'only' angry or frustrated, but did so when they felt annoyed (primary emotion) and when they thought that they do not want to hear or see anything about that issue anymore (primary cognition).

Turning to the two types of issue fatigue, the results showed similarities and differences in their associations to subsequent reactions. Concerning the commonalities, both were associated with the active and conscious avoidance of the issue in traditional media and interpersonal discussions. Referring back to the democracy-related arguments presented at the beginning of this paper, issue fatigue can stand against one basic requirement of a functioning democracy: it diminishes or even erodes citizens' level of information about certain issues, and their vigilance toward the related political developments. This is particularly problematic for issues that will remain on the political agenda and do not have a 'natural ending'. Consequently, political decisions could be taken without having been legitimized by the citizens.

Concerning the differences between the two types, the fatigued-suspicious searched significantly more often for alternative perspectives about the issue outside of the mainstream news, while simultaneously avoiding information in traditional media. It is likely that they used social media as an 'alternative source'. With regard to the discussion around fake news, misinformation, and filter bubbles on social network platforms, the first reaction to this result is probably to feel worried. In a democracy, the traditional media serve as fourth estate, and control political processes by providing citizens with the reliable information that they need in order to form their opinions. However, if citizens avoid such reliable information provided by traditional media, and instead rely solely on 'alternative sources' found in social media or other online sources, they may be more prone to 'undemocratic content', such as misinformation, conspiracy theories, and radical or extreme positions. However, this argument reflects a US–European perspective, and may be valid only in countries with a predominantly free and balanced information base presented by traditional media—such as Germany (Reporters Without Borders 2022a). For other countries, the interpretation is likely different. When compared to Germany, both Pakistan and Mexico score low on the 2022nd World Press Freedom index (Reporters Without Borders 2022b). In Mexico, traditional media is controlled by only a small number of entities; as a result, the interests of media companies and political parties are often interwoven, making the independence of the press, at its best, questionable (Reporters Without Borders 2022c). Nonprofit networks, such as (CONNECTAS 2022), aim to facilitate the production of independent, quality news. The results of our data showed that for 'war in Syria', Mexicans turned significantly more often to alternative sources when compared to Germans. Considering the characteristics of both media systems, we can conclude that issue fatigue is likely beneficial for the development of an informed and vigilant citizenry in Mexico, and might bring citizens to actively search for more independent perspectives than presented in the controlled traditional media. Interestingly, Pakistanis, when compared to Germans, turned significantly less often to alternative sources of news. Pakistan is clearly in a democratic transition (The Express Tribune 2013) and, despite the remaining problems with the low level of press freedom, the Pakistani media landscape is considered the most vibrant in Asia, presenting a wide range of news and opinions (Freedom House 2022; Reporters Without Borders 2022d). Considering these recent developments and transitions, we assume that the differentiation between 'traditional media' and 'alternative media' is not so clear in Pakistan and that, in consequence, our operationalization may have been blurry for Pakistani participants.

In contrast to the fatigued-suspicious, the fatigued-suffering-from-all unfollowed social media sources that disseminated related-issue information, and expressed their fatigue either on social media or in interpersonal discussions. In comparison to the fatigued-suspicious, the fatigued-suffering-from-all experienced more emotions, namely high-arousal negative emotions, as well as sadness and boredom. We assume that this emotional type of issue fatigue caused recipients to act 'in the heat of the moment' and use social media or interpersonal communication as a way to express their annoyance— either by deleting accounts from their news stream and/or by informing others about their issue fatigue. We find this specific combination worrying. In contrast to traditional media, in which recipients still incidentally come into contact with a fatigue issue, deleting information sources on social media is a rather severe reaction. In doing so, recipients can completely cut off an issue, unless they start to follow the relevant sources again. Furthermore, by additionally expressing their fatigue, recipients likely 'infect' others, by making them aware that a topic is bothersome. This could bring others, as well, to modify their amalgam of news sources on social media in a way that strictly cuts off more information on that topic, and both Pakistani and Mexicans showed higher tendencies for such reactions, compared to Germans. These countries are high-context collectivistic cultures (Hall 1990; Hofstede et al. 2010), in which close social relations are more important than in low-context individualistic cultures (such as Germany). As such, it is likely that the outlined problems are particularly relevant for such cultures.

### 9. Limitations and Future Research Directions

To our knowledge, our study is the first to show insights into different types of issue fatigue, as well as their associations with information avoidance and extended coping. That said, our approach could be improved and/or extended. Most prominently, our research was limited to two issues exclusively, and we cannot be sure whether we would replicate similar results when studying different issues. As such, we propose that future work consider additional issues. Moreover, as stated above, 'Trump' is not necessarily an issue per se, but a person. Even if we know from our topic-detection questionnaire what people associate with 'Trump' as a topic, the underlying mechanisms of how recipients define for themselves what a topic is, remain unclear. Therefore, we propose that future research also aim for a theoretically driven selection of issues, e.g., based on schema-theory (Eichhorn 1996).

Another issue is that, due to considerations of space, our study did not measure the characteristics of the recipients. However, we assume that a particular person's news consumption habits or their feeling of a "duty to keep informed" (McCombs and Poindexter 1983) could moderate or mediate the associations between issue fatigue and information avoidance or extended coping.

Moreover, our sample was rather homogenous for education. Consequently, our results refer to a group of formally highly educated individuals, limiting our ability to draw any conclusions for formally less educated people.

Lastly, from a theoretical perspective, issue fatigue is a phenomenon developing over time, and it is likely that levels of issue fatigue vary during a period of news coverage of an issue. For example, people experiencing issue fatigue could 'start' as the fatigued-suffering-from-all and, after a certain time, emotionally resign and shift to the fatigued-suspicious; this would also explain why we could find no cluster equivalent to the fatigued-suffering-from-all for 'Syria', as this issue has been far longer on the media agenda when compared to 'Trump'.

However, our cross-sectional study did not allow for studying any developments or changes in issue fatigue and subsequent reactions. For future research, we propose a panel-design, to study how issue fatigue arises and vanishes, and how these changes are associated with information avoidance and extended coping. For this, future research should also consider potential reasons for issue fatigue. In particular, the respective news coverage should be considered. It is likely that certain journalistic coverage—or at least how recipients perceive and evaluate it—could cause issue fatigue. If this should prove to be true, more knowledge about good journalistic practice with regard to extreme-lasting issues would be needed. We propose a mixed-methods design, combining objective measurements from a content analysis with the recipient's subjective perceptions and evaluations of news coverage. We argue that such a mixed-methods panel design would allow for the understanding of issue fatigue development, with its causes and consequences, in a more holistic way.

**Funding:** This research received no external funding.

**Institutional Review Board Statement:** The study conformed to the standard ethical principles. All respondents gave their consent to participate in the study and the data were hold anonymously. Ethical review and approval were waived, as the study did not involve sensitive personal data.

**Informed Consent Statement:** Informed consent was obtained from all subjects involved in the study.

**Data Availability Statement:** Not applicable.

**Conflicts of Interest:** The author declares no conflict of interest.

## Note

[1] The first part of the quote is taken from Magdalena Saldana from a panel discussion titled "The Comparison Trap? Current Theoretical and Methodological Challenges in Comparative Journalism Research," as indicated by Claussen (2020, p. 4).

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
