# Peer review of "When News Topics Annoy—Exploring Issue Fatigue and Subsequent Information Avoidance and Extended Coping Strategies"

_journalmedia, doi:10.3390/journalmedia3030037_

Round 1
Reviewer 1 Report
1. Line 15-can be changed to 'Recipients of the cognitive types turned to alternative sources of information...'
2. Line30, Break the sentence into two 'News outlets...'
3. Line 36, sentence 'Recipients process infor...' does not make sense
4. Line 80, delete 'a' before phenomenon
5. Line 144, 'Research questions' instead of 'Questions'
6. Line 153, 166, 193, "Western world"
7. Line 166, 'research questions' instead of 'question'
8. Line 173, 'first step', not 'first steps'
9. Line 235, if you use APA style, make sure you have proper in-text citation for interview quotes
10. Line 387, need a comma
11. Line 567, proper citation for website link
12. References-use proper APA style citation. For instance, Line 661, italicize 'Journal of Communication, 51'
Author Response
Dear reviewer,
in reflection of your comments, we have made several edits to our paper that we hope address all of the concerns raised. We sincerely thank you for your detailed feedback and are very grateful for your valuable hints for revising our paper. Below, we adress each of your concerns in a table.
Reviewer's comment | Author's response |
1. Line 15-can be changed to 'Recipients of the cognitive types turned to alternative sources of information...' | 1. Thank you. We changed line 15 accordingly. |
2. Line30, Break the sentence into two 'News outlets...' | 2. The sentence was broken into two. |
3. Line 36, sentence 'Recipients process infor...' does not make sense | 3. Thank you for pointing us to that. The sentence was changed to:
"Moreover, intensive coverage motivates recipients to process information more deeply (Ciuk & Yost, 2016), to increase their knowledge about such topics (Matthes, 2005) and even to engage in more political behavior such as signing petitions or voting (Weaver, 1991). " We hope, it becomes clearer now. |
4. Line 80, delete 'a' before phenomenon | 4. Done. |
5. Line 144, 'Research questions' instead of 'Questions' | 5. Done. |
6. Line 153, 166, 193, "Western world" | 6. Done. |
7. Line 166, 'research questions' instead of 'question' | 7. Done. |
8. Line 173, 'first step', not 'first steps' | 8. Done. |
9. Line 235, if you use APA style, make sure you have proper in-text citation for interview quotes |
9. Thank you very much for this comment. However, to be honest, we are not rather sure what to change in terms of APA7th quotation for interview quotes. We checked several websites considering this (see below), but we cannot find our mistakes. All quotes are shorter than 40 words and for all quotes we indicated that they are coming from an interview participant. In case of any misunderstandings, we kindly ask you to guide us in how to revise the manuscript. Thank you very much. Sources checked:
|
10. Line 387, need a comma | 10. Thank you. We now set the comma as follows: With a mean age of 27 years (SD = 10.0, MIN = 17, MAX = 70), participants belong mostly to the younger parts of society with the youngest participant being 17 years old. |
11. Line 567, proper citation for website link | 11. Thank you for pointing us to that. Again, we checked the aforementioned websites to correct the citation. What was somewhat difficult is that the page we cite has changing pieces (it is a "collection" of updated reports) and therefore no stable title. As such, we put in the reference list as title the "Investigaciones de Mèxico" as indicated in the heading of the tab in the browser (this element does not change). We think that this solution makes it the easiest for potential readers and hope that this solution finds your acceptance. |
12. References-use proper APA style citation. For instance, Line 661, italicize 'Journal of Communication, 51' |
12. Thank you for that hint. We apologize that we missed to italicize the necessary section in the reference list. We corrected that. We have taken the liberty of omitting the tack-changes for this, as this made the document rather unclear. We hope that this finds your acceptance. Additionally, we cross-checked the entire reference list and corrected further mistakes. |
Reviewer 2 Report
A well-written and interesting research article that the writer of these words believes should be published. The methodological and analytical part of the article deserves special attention. Both the research design itself and its implementation are in line with the canons of the art. The shortcomings of the article include above all the theoretical part, which is not very developed, and the apparent depth of the data. I would recommend the author/author to deepen the analyses with less obvious correlations. The literature used is up-to-date and recognised in the scientific world
Author Response
Dear reviewer,
we sincerely thank you for your constructive feedback and are very grateful for your valuable remarks on our paper. Reading your review, we understand that you generally evaluate the manuscript positive but that you recommend to reflect on two shortcomings:
- the theoretical base could be more substantiated
- the results section could be less detailed
We hope that this is correct.
Below, we answer your concerns:
- When reading our paper again, we shared your impression on "1. theoretical base" that the theoretical portion of the manuscript could be further developed. We therefore revised chapter 3 on the definition of issue fatigue. Our main focus was to substantiate our paragraph on the primary emotion and cognition by adding theoretical arguments for their occurence. As such, we do not "simply" rely on existing empirical results but reflect on the theoretical underpinnings of the findings.
- We see your point and agree that the empirical part of the paper is "dense". However, we were not to 100% sure if we correctly understood what you meant with: "deepen the analyses with less obvious correlations". Given that research on issue fatigue is still in its infancy and that the concept is not well studied yet, we see the necessity to sound empirical perspectives in order to grow knowledge. As such, we did not change the empirical portion of the paper. But of course we are open to further discuss your point with you!